# A Novel Triplet of Alisertib Plus Ibrutinib Plus Rituximab Is Active in Mantle Cell Lymphoma

**DOI:** 10.3390/cancers16244257

**Published:** 2024-12-21

**Authors:** Baskaran Subramani, Patrick J. Conway, Aisha Al-Khinji, Kun Zhang, Ritu Pandey, Daruka Mahadevan

**Affiliations:** 1Division of Hematology/Oncology, Department of Medicine, Mays Cancer Center, University of Texas Health San Antonio, San Antonio, TX 78229, USA; subramani@uthscsa.edu (B.S.); patrick.conway@keiseruniversity.edu (P.J.C.); aalkhinji@arizona.edu (A.A.-K.); zhangk@uthscsa.edu (K.Z.); 2Graduate School of Biomedical Sciences, University of Texas Health San Antonio, San Antonio, TX 78229, USA; 3Clinical Translational Science Program, University of Arizona, Tucson, AZ 85721, USA; 4Department of Cellular and Molecular Medicine, University of Arizona Cancer Center, Tucson, AZ 85721, USA; ritu@arizona.edu

**Keywords:** Aurora-A, Bruton tyrosine kinase, rituximab, mantle cell lymphoma

## Abstract

The mantle cell lymphoma field has advanced significantly due to ongoing developments in molecular pathogenesis prediction and novel therapeutic approaches. The area has undergone a revolution with the development of Bruton tyrosine kinase inhibitors, which are now the cornerstone of R/R MCL therapy. Developing and implementing new targeted and combination approaches has already improved therapeutic options, especially for refractory or relapsed diseases.

## 1. Introduction

Mantle cell lymphoma (MCL) is a subtype of B-cell non-Hodgkin lymphoma (NHL) characterized by a generally aggressive and heterogeneous clinical course, repeated remission, relapse, and resistance to therapy. A hallmark of MCL is t(11;14) translocation that leads to constitutive expression of cyclin D1, enhancing cell cycle transition from G1 to S phase [1] by activating CDK4/6 and promoting tumor cell proliferation. Among all aggressive B-NHLs, MCL patients have a worse prognosis overall [2], with long-term survival being 4.1 years for the high-risk group and 7.7 years for the low-risk group. MCL consists of several phenotypes of varying aggressiveness, which are classical, blastoid, and pleomorphic, with disparate responses to FDA-approved therapies [3].

The Bruton tyrosine kinase (BTK) inhibitors are FDA-approved as single agents in r/r MCL in the second (covalent inhibitors) or third line (non-covalent inhibitors) and are a backbone for combination with investigational targeted therapies for relapsed/refractory (r/r) MCL. With ibrutinib, the first-in-class covalent BTK inhibitor in r/r MCL (n = 111) patients demonstrated an objective response rate (ORR) of 68% with a complete response rate (CR) rate of 21% and a partial response rate of 47% [4]. Following the success of the covalent BTK inhibitor, ibrutinib, in second-line therapy (now discontinued), acalabrutinib and zanubrutinib (covalent BTK inhibitors) were developed and subsequently approved by the FDA for r/r MCL in the second line for their higher selectivity and decreased off-target effects, achieving overall response rates of 81% and 83%, respectively [5,6]. However, drug resistance, e.g., C481 binding site mutation and/or dysregulated PI3K/Akt/mTOR, is acquired with chronic therapy of covalent BTK inhibitors [7,8,9,10]. To overcome the C481 binding site mutation to covalent BTKi, non-covalent BTKi, pirtobrutinib, achieved an ORR of 57.8% and a CR of 20% in relapsed/refractory MCL after two lines of therapy [11]. However, the use of pirtobrutinib also faces the challenges of acquired non-C481 pathogenic mutations that maintain BCR signaling by BTK-dependent or BTK-impaired mechanisms, e.g., activation of the PI3K/AKT signaling pathway [12,13].

To overcome BTKi resistance or insensitivity, several combination therapies based on BTK inhibitors were investigated. A phase Ib trial investigated the responses to escalating doses of palbociclib (CDK4/6 inhibitor) plus ibrutinib in 27 r/r MCL patients [14], with an ORR not superior to ibrutinib alone [14]. Rituximab plus ibrutinib was investigated in a phase 2 trial in r/r MCL with an ORR of 88% [15]. At the 4-year follow-up, 58% (29 of 50) of the patients achieved a CR with a PFS of 43 months (range, 1–48) and 3-year 54% PFS, with median overall survival (OS) not reached and a 3-year OS of 69% [16]. Ibrutinib plus rituximab is an active combination compared to ibrutinib plus palbociclib; however, there is room for further improvements in targeted therapies for r/r MCL. Despite the progress made with BTKi-based regimens and other novel therapies against MCL, including CAR-T [17], revlimid plus rituximab [18], and bortezomib-based therapy [19], a sizeable population remains incurable.

In this study, we examined the therapeutic efficacy of combining alisertib, an Aurora A kinase ATP-site inhibitor, with ibrutinib or ibrutinib plus rituximab in two MCL cell lines. Aurora kinase A (AK-A) is a member of Aurora kinases (AKs), a family of serine/threonine protein kinases that play essential roles as regulators of mitosis [20,21,22,23,24,25]. Aberrant overexpression of AK-A is a poor prognostic marker in MCL [26]. AK-A inhibitors have shown impressive pre-clinical anti-tumor activity [26] as an active single agent to treat r/r aggressive B-NHLs [27] and peripheral T-cell lymphomas (PTCL) [28]. We hypothesized that dual inhibition of AK-A plus BTK by inhibiting mitosis and the chronic active B-cell receptor (BCR) signaling, respectively, would be synergistic as anti-MCL therapy by disrupting the proliferation of MCL cells insensitive to BTK inhibition and enhancing cellular apoptosis. We demonstrate that alisertib plus ibrutinib is synergistic in inhibiting MCL cells (Granta-519 and JeKo-1). In a Granta-519 cell-derived xenograft mouse model of MCL, a doublet of alisertib plus ibrutinib or a triplet of alisertib plus ibrutinib plus rituximab enhanced tumor growth inhibition, with improved overall survival.

## 2. Materials and Methods

### 2.1. Cells and Reagents

MCL cell lines: Granta-519 cells were obtained from Deutsche Sammlung von Mikroorganismen and Zellkulturen GmbH (Leibniz Institute DSMZ, Braunschweig, Germany). JeKo-1 cells were purchased from ATCC mycoplasma which had tested negative. The Granta-519 cell line authentication was performed using Promega PowerPlex16HS Assay at the University of Arizona Genetics Core and tested for mycoplasma contamination using a MycoAlert Mycoplasma Detection Kit (Lonza, Houston, TX, USA). Cell lines were maintained in RPMI 1640 medium with L-glutamine IX (Mediatech, Manassas, VA, USA) supplemented with 10% fetal bovine serum at 37 °C in a humidified atmosphere containing 5% CO_2_. Alisertib and ibrutinib were dissolved at 10 mM in DMSO as a stock solution and then diluted to the desired concentration for in vitro experiments, respectively. Rituximab was a gift from the University of Arizona Cancer Center clinic.

### 2.2. Cell Proliferation Assay

Granta-519 and JeKo-1 cells were cultured in 96-well culture plates seeded at 10,000 per well. Cells were allowed to grow for 24 h followed by the treatments designed with increasing concentrations of the testing agents for 4 days. Viable cell densities were determined using a CellTiter 96 Cell Proliferation Assay (Promega, Madison, WI, USA). Cell viability for each drug treatment against a control was determined with absorbance readings at 490 nm. The studies were performed in triplicates × 3, and IC50 values were calculated using Calcusyn 2.0 software (Biosoft, London, UK) and GraphPad Prism 9. For alisertib plus ibrutinib combinations, the equipotent ratio of the two drugs was calculated to determine the combination index (CI) to ascertain synergy, additivity, or antagonism, respectively. A control group was established for each drug treatment in six replicates. The effects of the combined treatments were determined by the combination-index (CI) and isobologram methods derived from the median-effect principle of Chou and Talalay [29] and GraphPad Prism.

### 2.3. Synergistic Analysis

The synergistic effect of alisertib with ibrutinib was examined using the median effect method, as originally described by Chou and Talalay [29]. The type of interaction between alisertib and ibrutinib is reflected by the combination index (CI) values. The CI < 1 indicates synergy; CI = 1, addictive; and CI > 1, antagonism. The combination index analysis was performed using Compusyn 1.0 software (combo-Syn, Inc., Paramus, NJ, USA).

### 2.4. Apoptosis Assays

Annexin V staining was used for the apoptosis assay. After being treated with Annexin V, the cells were harvested and rinsed with cold PBS once before being centrifuged for 5 min, cells were then suspended in 500 μL of 1× Annexin V binding buffer (BioVision, Mountain View, CA, USA, Annexin V-FITC Reagent Kit, Cat. #1001–1000), and then, 5 μL of Annexin V-FITC and 5 μL of propidium iodide (BioVision, Annexin V-FITC Reagent Kit) were added. After incubation for 5 min at room temperature in the dark, the samples were analyzed by flow cytometry.

### 2.5. Immunoblotting

MCL cells were lysed in lysis buffer to a final concentration of 1× RIPA buffer supplemented by Protease/Phosphatase inhibitor cocktail (Cell Signaling Technology, Danvers, MA, USA). Protein concentrations were determined using the BioRad protein assay kit (Hercules, CA, USA). A sample of 30 µg of protein was resolved by electrophoresis on a 10% SDS-PAGE, and the proteins were transferred onto nitrocellulose membrane and then incubated with 5% non-fat milk in TBST buffer (0.01 M Tris-Cl, 0.15 M NaCl, 0.5% Tween-20, pH 8.0) at room temperature for 1 h to block nonspecific binding. The proteins of interest were detected by subjecting the membrane to the respective antibodies and measured by a LI-COR Odyssey Infrared Imaging System. Antibodies used: Anti-Aurora B (CS#3094), anti-BTK (CS#8547), GAPDH (CS#2118) were purchased from Cell Signaling Technology (Danvers, MA, USA); anti-PI3K (SC#8010), anti-Bcl-2 (SC#7382), anti-C-Myc (SC#40), anti-NF-κB (SC#D14E12) and anti-p53 (SC#126) antibodies were purchased from Santa Cruz Biotechnology (Dallas, TX, USA).

### 2.6. Gene Expression Profiling

Granta-519 cells treated with alisertib 1.0 µM for 7 days to optimize polyploidy (3 biologic replicates) and untreated cells were grown for 7 days. Total RNA was extracted using TRIzol^®^ (Thermo Fisher Scientific, Waltham, MA, USA) from control and treated cells according to the protocol. RNA was checked using the Agilent 2100 Bioanalyzer (Agilent Technologies, Santa Clara, CA, USA). Affymetrix HuGene-2.0-st array (ThermoFisher, Waltham, MA, USA) was utilized to assess the differential expression profile in control vs. alisertib-treated Granta 519 cells (University of Tennessee Health Sciences, CITG). Probe summarization and normalization were performed using Affymetrix console 4.0 software. The ratio between treated and control cells was calculated for normalized expression data, and the results were filtered for genes that had difference of more than two-fold. An enrichment analysis was conducted for a list of genes that were differentially expressed. The Cluster Profiler package was used for gene ontology, and the ReactomePA package was used for pathway enrichment analysis. A *p* value <0.05, corrected for false discovery rate, was taken as significant. All analyses were carried out using R (v 3.4).

### 2.7. MCL Xenograft Model

Animal care and treatment were performed at the University of Tennessee Health Science Center in Memphis, TN, USA (Protocol #22-0321). SCID mice were injected subcutaneously with 1 × 10^7^ cells of Granta-519 cells into the right hind flank. Animals were divided into two groups: an experimental arm and a control group; the experimental group was further divided into four groups with twelve mice per cohort. When tumors reached a volume of 100 mm^3^, mice were divided into 5 groups: control group (saline), ibrutinib (10 mg/kg oral QD × 3 weeks), ibrutinib (10 mg/kg oral QD × 3 weeks) plus alisertib (30 mg/kg PO QD × 3 weeks), ibrutinib (10 mg/kg oral QD × 3 weeks) plus rituximab (10 mg/kg IV, once/week, × 4), and ibrutinib (10 mg/kg oral QD × 3 weeks) plus alisertib (30 mg/kg PO QD × 3 weeks) plus rituximab (10 mg/kg IV, once/week, ×4). The length and width of the subcutaneous tumors were measured with a caliper, and the tumor volume (TV) was calculated as TV = (L × W2)/2. Three mice/cohort were sacrificed at the end of treatment and nine mice at the end of study, or if they reached >2500 mm^3^ at any time during the study. Survival data were measured from the date of the injection to the end of the study.

### 2.8. Statistical Analysis

An estimation of tumor growth for each mouse was performed by fitting the least squares regression line of the tumor volume by day. The linearity was derived from the cube root of the observed tumor volumes. The slope of the regression line measured the tumor growth rate. The overall treatment effects on tumor growth inhibition were assessed using analysis of variance. Pair-wise differences between the groups adjusted for multiple comparisons were further analyzed with Tukey’s studentized range test. Analysis was performed using Prism (Graphpad, La Jolla, CA, USA). All *p*-values ≤ 0.05 were considered statistically significant. The Kaplan–Meier method was used for survival, and *p*-values were determined by log-rank analysis.

## 3. Results

### 3.1. Co-Inhibition of AK-A Plus BTK Is Synergistic in Inhibiting Cell Viability in MCL

In the cell proliferation inhibition assay for cell viability in MCL cell lines, Granta-519 and JeKo-1 treated with alisertib showed IC50 values of 23.7 nM (Figure 1A) and 3.01 nM, respectively (Figure 1D). In contrast, ibrutinib inhibited Granta-519 cell proliferation with an IC50 14.35 µM (Figure 1B) and JeKo-1 cells with an IC50 of 0.56 µM (Figure 1E). In the presence of 0.95 µM of ibrutinib, the alisertib cytotoxicity curve shifted to the left when the IC50 was lowered to 12.5 nM in Granta-519 cells (Figure 1C), showing the synergy of the two drugs. In contrast, in the presence of 0.56 µM of ibrutinib, the IC50 of alisertib in JeKo-1 cells was 4.2 nM (Figure 1F) versus 3.01 nM in the absence of ibrutinib. However, combination index analysis suggests that ibrutinib and alisertib were synergistic on both Granta-519 and JeKo-1 cells (Appendix A). Apoptosis analysis by cell cycle analysis showed that, in Granta-519 cells, alisertib plus ibrutinib progressively increased apoptosis over ibrutinib alone at 48 h, 72 h, 96 h, and 7 days (Appendix A).

### 3.2. Alisertib Inhibits Expression of Proteins in Chronic Active BCR Signaling

Western blotting of alisertib-treated Granta-519 (Figure 2A) and Jeko-1 (Figure 2B) MCL cells showed increased protein levels of AK-B and p53, likely a response to impending apoptosis. In contrast, decreased levels of PI3K, BTK, c-Myc, and Bcl-2 with increasing doses of alisertib ameliorated signaling through the chronic active BCR pathway. This is indicative of MCL cells’ impending loss of viability. AK-A inhibition in both ibrutinib-insensitive Granta-519 and ibrutinib-sensitive Jeko-1 cells amplified the effect on NF-kβ inhibition (Figure 2), a final common pathway of BTK inhibition, promoting apoptosis and inhibiting cell proliferation.

### 3.3. Alisertib Decreases the Level of Several Phospho-Proteins Involved in Cell Proliferation

Alisertib treatment (1 µM) for 1 week caused a significant decrease in phosphorylated RSK1/2/3 and Hck (*p* = 0.01) and a modest reduction in p38 phosphorylation (*p* = 0.068) (Figure 3). In contrast, none of the p53 phosphorylation sites were affected by alisertib; hence, DNA damage response (p53 Ser-15 phosphorylation), transcription-dependent and -independent apoptosis (Ser-46 phosphorylation), and mitochondrial translocation- and transcription-independent apoptotic function were unaffected.

### 3.4. Alisertib Induces Differential Gene Expressions in Gene Enrichment Analysis of MCL

Gene expression profiling of the alisertib-treated versus control cells showed differential expression of 764 genes, of which 523 genes (68%) were upregulated and 221 genes (32%) were downregulated. The enrichment analysis of pathways for upregulated genes upon alisertib treatment revealed changes in interleukin signaling followed by PI3K/AKT (Figure 4A). Many interleukins and cytokines that arose due to the response to cell stimulus were upregulated, and these bound to cell surface receptors to initiate signaling responses. Enrichment of genes that belonged to the MAPK pathway was not found. PIK3R3 (phosphoinositide-3-kinase regulatory subunit 3), one of the regulatory subunits of PI3K, was upregulated and had a role in multiple signaling activity driven by stimulation of receptors. FGF2 (fibroblast growth factor 2), CD28, and TRAT1, which also triggered PI3K activity, were upregulated too. But there was no evidence of AKT signaling pathway activation. There was a change in genes that were part of the PI3K/AKT signaling process, but there might have been a disruption in signaling due to phosphatases. Gene ontology enrichment analysis showed differential effects on cytokine and chemokine activity, as well as their receptors and cell adhesion molecules (Figure 4B). This confirms the enrichment of the same molecules that we detected in pathway analysis. Regarding the list of downregulated genes, we found that amino acid and organic acid biosynthetic processes were affected (Figure 4C).

### 3.5. Co-Administration of Alisertib Plus Ibrutinib with/Without Rituximab Is Synergistic in a Granta-519 Cell-Derived Mouse Model of MCL

In vivo data with Granta-519 cells in a SCID mouse xenograft model showed modest tumor growth inhibition (TGI) with alisertib (~13%) [30]. However, our in vitro data showed that ibrutinib was not an active drug in cell culture, but, when combined with alisertib, it showed significant inhibition of cell proliferation with enhanced apoptosis. We evaluated the anti-tumor activity of these kinase inhibitors in an SCID mouse xenograft model of MCL (Granta-519). Treatments with ibrutinib alone, ibrutinib plus alisertib (MLN), ibrutinib plus rituximab, and ibrutinib plus alisertib (MLN) plus rituximab were compared to each other and a vehicle control. Five cohorts of twelve mice in each arm were evaluated for tolerability (body weight), response (tumor growth inhibition), and survival. Mice treated with ibrutinib alone showed ~50% TGI, which contrasts with alisertib ~13% [13]. The combination of alisertib plus ibrutinib increased TGI to ~62%, which is additive. The doublet ibrutinib plus rituximab was better, at ~76% TGI. Finally, the triple combination of alisertib plus ibrutinib plus rituximab showed a TGI of ~82% (Figure 5A). Kaplan–Meier survival indicated a significant difference between the control vs. doublet- and triplet-treated mice (*p* < 0.0001) and ibrutinib- vs. combination-treated mice (*p* < 0.0001). There was no significant difference in survival between the different combinations (*p* ≥ 0.589) vs. ibrutinib or control (Figure 5B). All treatments were well tolerated, with no significant changes to body weight throughout the study (Appendix A).

## 4. Discussion

Novel combination therapies are needed for a majority of MCL patients failing first- and second-line therapies. Resistance to ibrutinib and second-generation BTK inhibitors develops within ~12 months [14,31] due to acquired mutations in the kinase domain, downstream kinases, and heterogeneity in chronic active BCR signaling in MCL. A consensus view in the treatment of MCL involves combinational therapies that target multiple signaling pathways. Alisertib, an AK-A inhibitor that causes cancer cell arrest in the G2/M phase [32,33], would conceivably be in tandem with a BTK inhibitor that targets cell growth to restore the response to BTK inhibitors. Our study showed that alisertib and ibrutinib administered together achieved therapeutic synergy on Granta-519 MCL cells insensitive to BTK inhibition (Figure 1A–C). In contrast, while JeKo-1 cells are sensitive to ibrutinib [34] and alisertib, respectively (Figure 1D,E), co-administration of the two drugs led to less pronounced inhibition, although it was synergistic (Figure 1F). While the mechanisms for differential responses are beyond the scope of this study, we show that the Jeko-1 cells’ sensitivity to alisertib led to a significant reduction in the BTK protein (Figure 2). There is evidence suggesting that reduced BTK expression and upregulated PI3K pathway and BCL-2 signaling due to chronic ibrutinib treatment is a mechanism of resistance [35,36]. Western blot analysis showed that alisertib downregulated c-Myc expression, a positive regulator of the chronic active BCR pathway [19], with significantly decreased expression of PI3K, BTK, BCL-2, and NF-kβ (Figure 2) in both MCL cell lines. The phospho-protein array of Granta-519 cells treated with alisertib showed reductions in several kinases, including p38, RSK, and HCK (Figure 3). These kinases are the integral components of the MAPK and SRC signaling pathways. The addition of ibrutinib to alisertib enhanced apoptosis in a time-dependent manner compared to ibrutinib alone in Granta-519 MCL cells (Appendix A). The gene expression analysis showed there was upregulation in the expression of genes in the PI3K/AKT pathway and cell surface receptors, but we did not see activated AKT signaling, suggesting that there is an interplay of phosphatases and kinases inhibiting specific signaling processes.

Previously, we showed in a pre-clinical model [37,38] and in a clinical trial [39] that combining alisertib with vincristine plus rituximab (MVR) is synthetically lethal in aggressive B-NHL, including MCL. Here, we replaced vincristine with ibrutinib and evaluated the effects of alisertib plus ibrutinib, ibrutinib plus rituximab, and alisertib plus ibrutinib plus rituximab, respectively, in a Granta-519 MCL cell-derived xenograft mouse model. In the mouse model, ibrutinib demonstrated higher anti-tumor activity compared to alisertib [20,40]. The doublets of ibrutinib plus alisertib or ibrutinib plus rituximab showed similar tumor growth inhibition; however, the triplet of alisertib plus ibrutinib plus rituximab showed an augmented anti-tumor response (Figure 5A). The doublet and triplet ibrutinib combinations demonstrated similar survival, at 85–95%, compared to ibrutinib alone, which was <50% at day 16 (Figure 5B).

## 5. Conclusions

In conclusion, our study suggests that alisertib has multidimensional effects, suppressing numerous proteins in the cell proliferative signaling pathways; alisertib and ibrutinib are synergistic, treating ibrutinib-insensitive MCL. A B-cell receptor (BCR)-related gene signature may be a predictive biomarker for the therapeutic response to alisertib and ibrutinib combination.

## Figures and Tables

**Figure 1 cancers-16-04257-f001:**
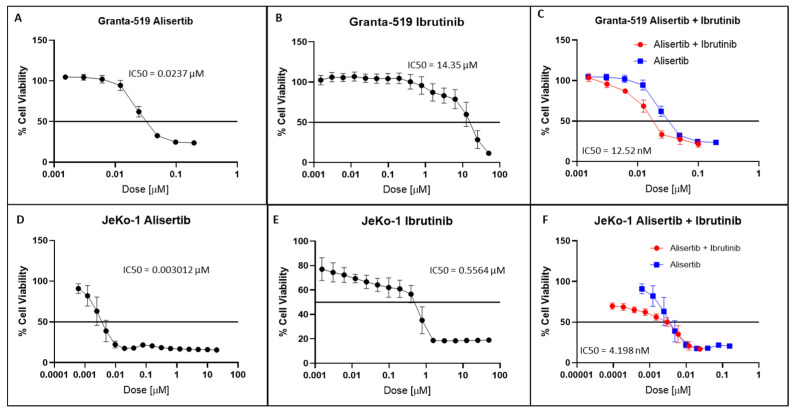
MTS cell proliferation assay of MCL cell lines ((**A**–**C**) Granta-519; (**D**–**F**) JeKo-1). (**A**) IC50 of alisertib = 23.7 nM. (**B**) IC50 of Ibrutinib = 14,350 nM. (**C**) Leftward shift of concentration–effect curve for alisertib plus ibrutinib indicates significant synergism with an IC50 = 12.52 nM. (**D**) IC50 of alisertib = 3.01 nM. (**E**) IC50 of ibrutinib = 556.4 nM. (**F**) Ibrutinib and alisertib are synergistic in JeKo-1 inhibition.

**Figure 2 cancers-16-04257-f002:**
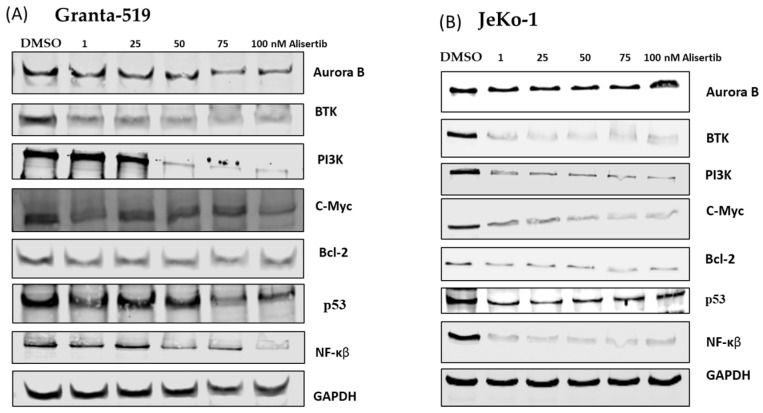
(**A**) Granta-519 cells and (**B**) Jeko-1 cells treated with alisertib in DMSO, 0, 1, 25, 50, 75, and 100 nM alisertib for 24 hrs. Western blotting of these cells showed decreased protein levels of BTK, PI3K, c-Myc, NF-kβ, and Bcl-2 in a dose-dependent manner. Quantification of signal intensity of the blots (Appendix A Granta-519 and Appendix A Jeko-1, Appendix A have the uncropped Western blots).

**Figure 3 cancers-16-04257-f003:**
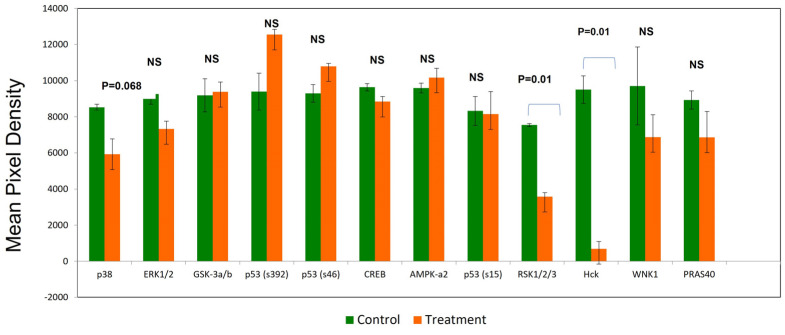
Phospho-protein array of Granta-519 cells treated with alisertib 1.0 μM for 7 days to optimize polyploidy showed no significant changes in p53 phosphorylation at Ser-15, Ser-46, or Ser-392. Of the kinases, decreased phosphorylation with alisertib was observed with HCK > RSK 1/2/3 > p38. Phospho-protein array is displayed as a percentage of the pixel density from 2 biologic replicates.

**Figure 4 cancers-16-04257-f004:**
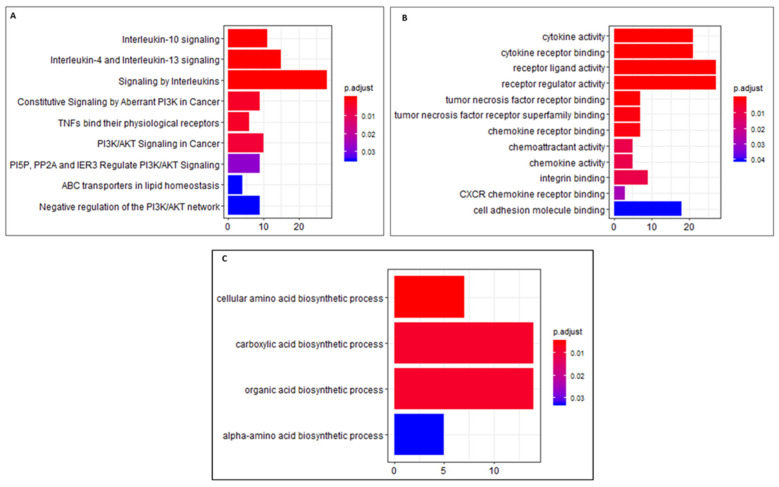
Gene enrichment analysis of differentially expressed genes for pathways and molecular function pre-/post-alisertib treatment of Granta-519 (BCR-low) cells. (**A**) Altered signaling pathways by upregulated genes. (**B**) Gene ontology molecular functions for upregulated genes. (**C**) Downregulated cellular processes.

**Figure 5 cancers-16-04257-f005:**
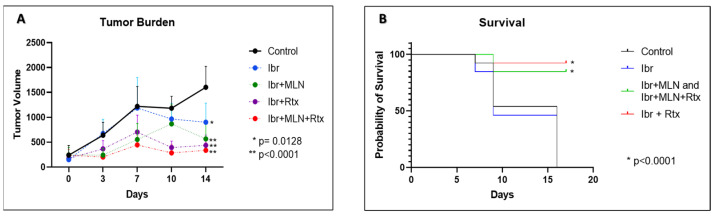
Granta-519 xenograft SCID mouse model of MCL. (**A**) Tumor growth inhibition: N = 12 mice per cohort treated with ibrutinib, ibrutinib plus alisertib, ibrutinib plus rituximab, and triple therapy versus control. On day 14, the differences between the tumor burdens of the control group and each of the treatment groups were statistically significant (GraphPad Prism version 8.4.1). (**B**) Kaplan–Meier survival curves of control vs. ibrutinib vs. ibrutinib + alisertib vs. ibrutinib + rituximab and ibrutinib + alisertib + rituximab. There was no significant difference between control and ibrutinib-treated mice (*p* = 0.6258). There was a significant difference between the control and doublet- and triplet-treated mice (** *p* < 0.0001) and between the ibrutinib- and combination-treated mice (* *p* < 0.0001). There was no significant difference between the different combination treatments (*p* ≥ 0.589). *p*-values were determined by log-rank analysis.

## Data Availability

The original contributions presented in the study are included in the article/Appendix A; further inquiries can be directed to the corresponding author.

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
