# Peer review of "A Novel Triplet of Alisertib Plus Ibrutinib Plus Rituximab Is Active in Mantle Cell Lymphoma"

_cancers, 2024, doi:10.3390/cancers16244257_

Round 1

Reviewer 1 Report

Comments and Suggestions for Authors

The manuscript aims to develop a combination therapeutic strategy for mantle cell lymphoma, targeting Aurora A Kinase with alisertib, BTK with ibrutinib, and CD20 with rituximab. While this triple targeting is interesting, the manuscript is premature for publication due to the following issues: it is a descriptive study without in-depth mechanistic analysis, and it is based on two cell lines without analysis of primary cancer cells. Most data are weak and lack appropriate analyses, such as RNA-seq data. There is no cell cycle analysis, which is important to study alisertib .

Comments on the Quality of English Language

English is acceptable.

Author Response

Comment: The manuscript aims to develop a combination therapeutic strategy for mantle cell lymphoma, targeting Aurora A Kinase with alisertib, BTK with ibrutinib, and CD20 with rituximab. While this triple targeting is interesting, the manuscript is premature for publication due to the following issues: it is a descriptive study without in-depth mechanistic analysis, and it is based on two cell lines without analysis of primary cancer cells. Most data are weak and lack appropriate analyses, such as RNA-seq data. There is no cell cycle analysis, which is important to study alisertib.

Response: we conducted gene expression profiling, reported the results, and discussed the results in the discussion (lines 307 - 310)

Alisertib has been widely documented to arrest cells at G2/M phase, we addressed this question in the discussion (Ref #22 and 23).

Reviewer 2 Report

Comments and Suggestions for Authors

The authors explore the effects of alisertib in a triple combination with ibrutinib and rituximab in two cell line models of mantle cell lymphoma with a different sensitivity to alisertib, both in vitro and in a murine model.

The paper is clearly organized and well written. Data appear convincing, although some data on upregulated genes remain descriptive, and their contribution to the biological effects observed in the study remain speculative.

The authors may discuss the article “Drug Repurposing Screen Identifies Novel Classes of Drugs with Anticancer Activity in Mantle Cell Lymphoma” by HanC et al in Combinatorial Chemistry & High Throughput Screening, Volume 22, Issue 7, 2019. Pages 483-495

Author Response

Comment 1: The paper is clearly organized and well written. Data appear convincing, although some data on upregulated genes remain descriptive, and their contribution to the biological effects observed in the study remain speculative.

Response: We further discussed our findings of gene enrichment analysis in the discussion (lines 307 - 310).

Comment 2: The authors may discuss the article “Drug Repurposing Screen Identifies Novel Classes of Drugs with Anticancer Activity in Mantle Cell Lymphoma” by HanC et al in Combinatorial Chemistry & High Throughput Screening, Volume 22, Issue 7, 2019. Pages 483-495

Response: we respectfully ask to provide a rationale for this request.

Reviewer 3 Report

Comments and Suggestions for Authors

This pre-clinical work of alisertib plus ibrutinib plus rituximab as active in Mantle Cell Lymphoma, is an interesting study

It is well performed and the manuscript is clear, and the documents in supplemental material is sufficient to cover the readers request.

I have nothing to add, only a minor typo at line 158 where the 1x107 cells should be correcter to 1x 107 cells

Author Response

Comment: a minor typo at line 158 where the 1x107 cells should be corrected to 1x 107 cells

Response: the typo was corrected.

Reviewer 4 Report

Comments and Suggestions for Authors

This study provided a novel combinatorial therapy for MCL. I have some comments about the study.

1. Combination index should be calculated to prove the synergy of ibrutinib and alisertib.

2. P53 does not increase in figure 2.

3. Caspase-3, caspase-8, and cleaved-PARP are better and more straightforward markers for demonstration of apoptosis.

4. The advantage of combination theprapies including alisertib in TGI and survival in the mouse models are not remarkable.

Author Response

Comments:

1. Combination index should be calculated to prove the synergy of ibrutinib and alisertib.

   Response: Analysis was conducted as requested and provided in Supplementary data Table 1.

 2. P53 does not increase in figure 2.

    Response: After careful review of the western blot and quantification for Granta-519, there is a non-significant increase in p53..

 3. Caspase-3, caspase-8, and cleaved-PARP are better and more straightforward markers for     demonstration of apoptosis.

Response:  Although caspase-3, caspase-8, and cleaved-PARP are valuable marker for apoptosis, we believe annexin V with flow cytometry is a standard method to measure apoptosis analysis, 

4. The advantage of combination therapies including alisertib in TGI and survival in the mouse models are not remarkable.

Response: The combination of alisertib and ibrutinib is tolerated and active. Increased dosing of both agents may further increase TGI and overall survival. These studies are planned.

Round 2

Reviewer 1 Report

Comments and Suggestions for Authors

The revision has improved the manuscript.

Author Response

Please refer to the attached document for the changes that have been implemented.

Reviewer 4 Report

Comments and Suggestions for Authors

 It will be better if scatter plots of flow cytometry can be provided for Figure S1

Author Response

(The authors gave the same response as above.)
